# Response to Bovine Viral Diarrhea Virus in Heifers Vaccinated with a Combination of Multivalent Modified Live and Inactivated Viral Vaccines

**DOI:** 10.3390/v15030703

**Published:** 2023-03-08

**Authors:** Shollie M. Falkenberg, Rohana P. Dassanayake, Lauren Crawford, Kaitlyn Sarlo Davila, Paola Boggiatto

**Affiliations:** 1Ruminant Diseases and Immunology Research Unit, National Animal Disease Center, USDA, Agricultural Research Service, Ames, IA 50010, USA; 2Department of Pathobiology, College of Veterinary Medicine, Auburn University, Auburn, AL 36849, USA; 3Infectious Bacterial Diseases of Livestock Research Unit, National Animal Disease Center, USDA, Agricultural Research Service, Ames, IA 50010, USA

**Keywords:** bovine viral diarrhea virus, virus neutralizing titer, cell mediated response, viral vaccines

## Abstract

Bovine viral vaccines contain both live or inactivated/killed formulations, but few studies have evaluated the impact of vaccinating with either live or killed antigens and re-vaccinating with the reciprocal. Commercial dairy heifers were utilized for the study and randomly assigned to three treatment groups. Treatment groups received a commercially available modified-live viral (MLV) vaccine containing BVDV and were revaccinated with a commercially available killed viral (KV) vaccine containing BVDV, another group received the same KV vaccine and was revaccinated with the same MLV vaccine, and yet another group served as negative controls and did not receive any viral vaccines. Heifers in KV/MLV had higher virus neutralizing titers (VNT) at the end of the vaccination period than heifers in MLV/KV and control groups. The frequency of IFN-γ mRNA positive CD4+, CD8+, and CD335+ populations, as well as increased mean fluorescent intensity of CD25+ cells was increased for the MLV/KV heifers as compared to KV/MLV and controls. The data from this study would suggest that differences in initial antigen presentation such as live versus killed could augment CMI and humoral responses and could be useful in determining vaccination programs for optimizing protective responses, which is critical for promoting lifetime immunity.

## 1. Introduction

Vaccination is an important component with respect to both BVDV control and limiting reproductive losses associated with BVDV infections. Vaccines licensed for use against BVDV are often multivalent, containing both viral and bacterial antigens, and are available in North America as either modified-live viral (MLV) or inactivated (i.e., killed) vaccines (KV) [1]. In general, MLV vaccines are thought to induce a balanced cell mediated and humoral response in comparison to killed vaccines [1]. Fetal protection (FP) has been demonstrated utilizing either MLV or KV vaccines. However, a greater number of licensed MLV BVDV vaccines have demonstrated FP claims, are generally considered to provide higher protection, and have a longer duration of immunity [2,3,4]. Despite evidence of the increased protection afforded by MLV vaccines, KVs continue to be widely used by cattle producers (NAHMS Beef 2007–2008) due to safety concerns associated with MLV vaccinations in close time proximity to breeding or during gestation [5]. Given the potential benefit of increased protection with MVL vaccines and the safety of KVs, different approaches need to be explored to better understand immune responses after initial priming with a live or killed antigen to inform MLV and KV vaccine programs. Previous studies comparing reproductive protection utilizing either a MLV or KV after primary vaccination with a MLV demonstrated an improvement in protection when incorporating a KV in the vaccination program [6]. Similarly, calves initially vaccinated with a MLV at two months of age, again at weaning, followed by a MLV or a KV 5 months later, reported that calves receiving the KV had higher titers than the calves only receiving a MLV [7]. While most recent studies have utilized priming with a MLV followed by a KV antigen, initial studies utilizing a combination approach were first vaccinated with KV vaccines and then boosted with a MLV vaccine [8,9]. Initial reports demonstrated increased serum neutralizing titers against BVDV when cows were revaccinated with a MLV vaccine after primary vaccination with a KV, and the authors suggested that a priming strategy with a KV followed by a MLV may achieve titers greater than observed with only utilizing either vaccine alone [9]. This was further expanded in BVDV efficacy studies that demonstrated a two-step vaccination protocol utilizing a MLV vaccine 4 weeks after administration of KV could provide fetal protection against BVDV infection [8,10,11]. Collectively, these data would suggest that improved protection can be achieved utilizing a combo MLV and KV approach rather than relying on only one vaccine alone and would indicate a benefit when combining live and killed antigens. Additionally, the timing between vaccination in each of the studies varied, suggesting the benefit of a combination approach may be associated the difference in antigen presentation of live and killed antigens. While previous data would support a combination approach, it is unknown if the initial presentation of live or killed antigens modulates or primes the immune system more effectively.

Initial priming of the immune response after initial antigen exposure may result in alteration of the immune profile, the magnitude, or the duration of the immune response, thereby altering subsequent responses when encountering the same antigen. The pathogen, site of replication, type of infection, and inflammatory signals, among other factors, contribute changes in the immune profile/response such as T helper 1 (Th1; cell mediated immune response) or a T helper 2 (Th2; antibody response) response [12]. In the context of vaccines, the type of adjuvant and the difference between MLV and KV can contribute to eliciting different immune profiles and responses [13]. Given the potential for immune modulation or priming as a result of initial antigen exposure, it is important to understand if the initial exposure to live or killed antigens alters the subsequent response to a live or killed antigen. The ability to use a combination approach to vaccination provides the opportunity to address safety concerns surrounding MLV vaccines and potential efficacy concerns with KV, but the combination that elicits optimal immunity is unknown. A better understanding is needed for the placement of live and killed vaccines in vaccination programs to optimize immunity and subsequent protection. Therefore, the goal of this research was to determine if immune response profiles differed depending on the sequence of live or killed antigens after administration of BVDV MLV or KV vaccines.

## 2. Materials and Methods

### 2.1. Animals and Sample Collection

Animals housed and samples collected at the National Animal Disease Center (NADC) were handled in accordance with the Animal Welfare Act Amendments (7 U.S. Code §2131 to §2156). All procedures were approved by the Institutional Animal Care and Use Committee of the NADC (ARS-2019-799).

Approximately one month prior to initiation of the study, twenty-four Holstein heifers (3–4 month of age), which tested negative for BVDV antigens, were delivered to the NADC. Upon arrival, all heifers were subjected to routine processing that included administration of prophylactic antibiotic (Draxxin, Zoetis, Inc., Parsippany, NJ, USA), pour-on dewormer, (Cydectin, Elanco Animal Health, Inc., Greenfield, IN, USA), oral dewormer, (Valbazen, Zoetis, Inc., Parsippany, NJ, USA), and a clostridial vaccine (Vision 8; Merck Animal Health, Madison, NJ, USA). All products were used as per manufacturers’ recommendations and administered approximately one month prior to administration of viral vaccines. Blood samples were collected from all heifers prior to the administration of the initial dose of viral vaccines. Heifers were randomly allocated to their respective experimental group and administered their respective vaccine or remained as non-vaccinated controls. Nine heifers (n = 9) were initially administered a commercially available pentavalent MLV vaccine (BoviShield Gold 5; Zoetis, Inc., Parsippany, NJ, USA) containing BVDV type 1 and type 2, bovine herpes virus-1 (BHV-1), bovine respiratory syncytial virus (BRSV), and parainfluenza type 3 virus (PI-3) per manufacturer recommended volume and route. Nine heifers (n = 9) were initially administered a commercially available multivalent KV vaccine (ViraShield 6, Elanco Animal Health, Inc., Greenfield, IN, USA) containing BVDV type 1 and type 2, BHV-1, BRSV, and PI-3 per manufacturer recommended volume and route. Six heifers (n = 6) remained as non-vaccinated controls. Blood samples were collected on weeks 2, 4, 8, and 12 post-initial vaccination for evaluation of virus neutralizing titers (VNT) and cell mediated immune (CMI) responses over time. After samples were collected at 12 weeks post-vaccination, heifers were revaccinated with MLV or KV as per the manufacturer recommended volume and route. Heifers that initially received the MLV vaccine (BoviShield Gold) subsequently received the KV vaccine (ViraShield 6) (designated as MLV/KV group) and inversely the heifers that initially received the KV vaccine (ViraShield 6) subsequently received the MLV vaccine (designated as the KV/MLV group). The same sampling schedule was followed, starting with the sample at 14 weeks post-initial vaccination or 2 weeks post-revaccination (14-2) and continuing with samples at 16-4, 20-8, and 24-12.

The vaccination and sample collection timing were chosen to allow adequate time for cell mediated immune (CMI) and humoral immune responses to develop and to evaluate the development of the CMI response. The timing was based on previously published studies that evaluated memory responses associated with BVDV [14,15,16,17,18].

### 2.2. BVDV Strains

Non-cytopathic (ncp) field strains BVDV-1a (PI34) and BVDV-2a (PI28) were selected to stimulate peripheral blood mononuclear cells (PBMC) and evaluate cell mediated recall responses. Cytopathic reference strains BVDV-1a (Singer) and BVDV-2a (296c) were used for virus neutralization assays. Details regarding complete genome sequencing and BVDV isolate characterization are previously described in the literature, in addition to GenBank accession numbers [18,19]. BVDV strains were propagated as previously described and according to standard protocol [20]. Viruses were titered through serial 10-fold dilutions with replicates of five wells per dilution on bovine turbinate (BTu) cells. Cytopathic effect (CPE) was evaluated for cytopathic refence strains, and the cell layer was fixed for non-cytopathic strains and stained according to standard immunoperoxidase staining protocol using the E2 protein-specific monoclonal antibody N2 and horseradish peroxidase-conjugated protein G for the ncp isolates [21].

### 2.3. Isolation, Culture, and Preparation of Peripheral Blood Mononuclear Cell (PBMC) for Flow Cytometry

Whole blood was collected via jugular venipuncture into ACD tubes (BD Vacutainer, Solution A). Peripheral blood was processed for isolation of PBMC using SepMate tubes (Stemcell Technologies, Cambridge, MA, USA) according to the manufacturer’s instructions and as previously described [17,18]. Cell pellets were resuspended in 10 mL of PBS and passed through a 40 µm filter to remove cell debris prior to determination and standardization of the total number of live PBMCs using the Muse™ Cell analyzer per manufacture’s recommendation and as previously described [18]. Adjusted cell suspensions were prepared as previously described [17,18]. Briefly, 100 µL of PBMC suspension containing ~1 × 10^6^ live cells were added to respective wells of a 96-well round bottom plate containing 100 µL of fresh complete cell culture medium composed of RPMI, supplemented as described previously. Cells were plated in duplicate for each respective non-stimulation or stimulation method for each animal. Plated cells were maintained at 37 °C in a humid atmosphere of 5% CO_2_.

### 2.4. Antigen and Mitogen Stimulation

As previously described [17,18], twenty-four hours after plating PBMC, 50 µL of media was removed from the respective antigen stimulation wells for each animal and replaced with 50 µL of each respective BVDV virus (BVDV-1a (PI34) and BVDV-2a (PI28)) at an approximate multiplicity of infection (MOI) of 1. Forty-eight hours after the cells were plated, 50 µL of media was removed from the wells designated for mitogen stimulation and 50 µL of eBiosciences cell stimulation cocktail (PMA/ionomycin; 8 µL diluted in 1 mL complete RPMI-1640) was added. Two wells were not stimulated over the course of the culture period and were used as non-stimulated controls as a measure of background responses. Approximately 1.5 h after the addition of the cell stimulation cocktail, all plated cells were prepared for surface and intracellular staining.

### 2.5. Preparation of PBMC for Flow Cytometry

At the end of the culture period, approximately 48 h post-isolation, 24 h post-BVDV stimulation, and 1.5 h post-mitogen stimulation, plates were centrifuged at 300× *g* for 4 min. Supernatant was discarded by flicking the plate in one swift movement. Duplicate wells for each respective animal, non-stimulation, and stimulation method were resuspended in 100 µL of PBS, then pooled into one well, for each stimulation condition. The pooled samples were transferred to a 96-well round bottom plate for further processing. Pooled cell pellets were pelleted by centrifugation at 300× *g* for 4 min and washed with PBS. This procedure was repeated two times. After the last wash step, the supernatant was discarded, pellets were resuspended and incubated with 100 μL (1:100 dilution) of the respective primary monoclonal antibody (mAb) mix in the dark at room temperature for 15 min for identification of PBMC subpopulations (Table 1). After two wash steps with PBS, the secondary antibody conjugate mix (100 µL at 1:100 dilution) was added to the respective wells by resuspending the pellet and incubating in the dark at room temperature for 15 min. After two wash steps with PBS, cells were further utilized in the PrimeFlow assay for CD4, CD8, BVDV, and IFN-γ detection and quantification as previously described [17,18].

### 2.6. Virus Neutralization (VN) Assay

VN assays were performed according to a previously described protocol [22] using the serum collected. Serial 2-fold dilutions of each antiserum in MEM were prepared, starting from a 1:2 initial dilution. In cell culture 96-well microplates, using replicates of three wells for each serum dilution, a 50-μL aliquot of diluted serum and a 50-μL aliquot of virus containing 100 TCID50 were added to each well and incubated for 1 h at 37 °C. At the end of the incubation period, primary BTu cells were added. This was accomplished by the addition of 20,000 BTu cells (in a 100-μL aliquot of MEM and 10% FBS) to each well. Microplates were incubated for 4 days at 37 °C in a 5% CO_2_ incubator. Replication of the virus was evaluated by observance of CPE in the wells. The neutralization results of the 3 wells at each respective serum dilution were evaluated for neutralization or lack of CPE. These results were used for the calculation of the endpoint titer using the Spearman–Kärber method, as previously described [23] and reported as the virus neutralization titer (VNT).

### 2.7. Data Analysis

The frequency of cells positive for the respective PBMC populations (CD4+, CD8+, CD25+ and CD335+) was calculated for each sample using FlowJo^®^ software (BD BioSciences, Franklin Lakes, NJ, USA). Within the cell population of interest, the frequency of positive cells for BVDV and IFN-γ was determined. The frequency (percent) of cells positive for IFN-γ mRNA was determined by subtracting the background expression in the non-stimulated cells from the stimulated cells. The mean fluorescent intensity (MFI) was also determined for CD25+ cells using FlowJo^®^ software. The VNT undetectable at 1:2 dilution (<2) was considered to be negative. Statistical analyses were performed using R version 4.2.0. The lm function was used to evaluate the effect of vaccination treatments on each variable for BVDV isolates used for stimulation using the model y = vaccine treatment + error within each timepoint. The package lsmeans was used to generate pairwise comparisons between vaccination treatment group means at each respective timepoint for each BVDV isolate used for stimulation using the Tukey HSD method. Means were declared significantly different at *p* < 0.05. All figures were generated using the R package ggplot2.

## 3. Results

### 3.1. BVDV-2a Results

Serological responses as determined by VNT were evaluated. The average VNTs for each treatment group against BVDV--2a (Figure 1) and BVDV-1a (Appendix A) reference strains were determined to be <64 prior to vaccination, suggesting that the initial administration of the vaccines was most likely in the presence of a maternal antibody from passive transfer given the age of the heifers at the time of vaccination. To determine a change in VNT after vaccination, an increase or decrease from the mean baseline VNT for each group and comparisons among treatment groups were used as the criteria. In the control group, BVDV titers declined and were undetectable (<2) by week 8 and remained undetectable (<2) for the remainder of the study period, suggesting these animals were not exposed to BVDV (Figure 1 and Appendix A).

An increase in BVDV-2a titers was observed for the MLV/KV group on week 4 post-vaccination, and a significant increase was observed on week 8 as compared to the control and KV/MLV groups (Figure 1). Post-revaccination week 2 and 4 (14-2 and 16-4), BVDV-2a titers remained greater for the MLV/KV group, compared to the control and KV/MLV groups (Figure 1). Additionally, peak BVDV-2a titers for the MLV/KV group were observed on week 16-4. Post-revaccination week 4 (16-4), an initial increase in titer was observed for the KV/MLV group, but a significant increase from the controls was not observed until week 20-8 (Figure 1). While the titers were not significantly different from the MLV/KV group, an increase in the BVDV-2a titer was observed for the KV/MLV group from week 16-4 through to the conclusion of the study (Figure 1). Similar to results observed for the BVDV-1a titers, cattle in the KV/MLV group had the higher titers, 2784 at the conclusion of the study, on the final sample time point (24-12) for BVDV-2a as compared to 2070.5 for the MLV/KV group (Appendix A).

The alpha chain of interleukin-2 receptor (IL-2α) or CD25 is considered a cellular activation marker and is expressed on the surface of activated T- and B-lymphocytes. After BVDV-2a PBMC stimulation, a significant increase in CD25 MFI was observed for the MLV/KV group from week 4 through week 12 post-vaccination compared to the control and KV/MLV groups (Figure 2). Post-revaccination, CD25 MFI was significantly increased in the MLV/KV group at all time points analyzed when compared to control animals. At week 2 post-revaccination, levels of CD25 were also significantly increased in PMBC from the MLV/KV as compared to the KV/MLV group (Figure 2). CD25 MFI was significantly greater in BVDV-2a stimulated PBMC from the MLV/KV and KV/MLV groups as compared to control from week 14-2 through to the conclusion of the study (Figure 2). Additionally, peak CD25 MFI for BVDV-2a stimulated PBMC for both the MLV/KV and KV/MLV groups was observed on week 20-8. The greatest CD25 MFI for BVDV-2a stimulated PBMC was observed from cattle in the KV/MLV group on week 20-8 (Figure 2).

As previously reported [18], an increased frequency of IFN-γ mRNA-positive cells was observed in PBMC stimulated with BVDV from vaccinated cattle (Figure 3, Figure 4 and Figure 5) as compared to non-vaccinated controls. After BVDV-2a PBMC stimulation, a significant increase in the frequency of IFN-γ mRNA-positive CD4+ cells was observed for the MLV/KV group from week 8 post-vaccination through week 16-4 post-revaccination compared to the control and KV/MLV groups (Figure 3). The peak frequency of IFN-γ mRNA-positive CD4+ cells after BVDV-2a stimulation was observed on week 16-4 post-revaccination and this peak was significantly greater than the KV/MLV and control groups (Figure 3). While not significantly different from other treatment groups, an increase in the frequency of IFN-γ mRNA-positive CD4+ cells after BVDV-2a stimulation was observed on week 16-4 post-revaccination and remained relatively consistent until the conclusion of the study for the KV/MLV group (Figure 3).

After BVDV-2a PBMC stimulation, a significant increase in the frequency of IFN-γ mRNA-positive CD8+ cells was observed for the MLV/KV group on weeks 8 and 12 post-vaccination compared to the control and KV/MLV groups (Figure 4). Post-revaccination, MLV/KV cattle had significantly higher frequencies of IFN-γ mRNA-positive CD8+ cells as compared to control animals at all time points analyzed (Figure 4). However, as compared to the KV/MLV group, the frequency of IFN-γ mRNA-positive CD8+ cells was only significantly higher at the last time point analyzed (24-12) (Figure 4). While not significantly different from other treatment groups, an increase in the frequency of IFN-γ mRNA-positive CD8+ cells after BVDV-2a stimulation was observed on week 16-4 post-revaccination and remained relatively consistent until the conclusion of the study for the KV/MLV group (Figure 4).

After BVDV-2a PBMC stimulation, a significant increase in the frequency of IFN-γ mRNA-positive CD335+ cells for the MLV/KV group on weeks 2-8 post-vaccination was observed when compared to control and KV/MLV groups (Figure 5). No significant differences were observed between the MLV/KV and KV/MLV groups from week 12 post-vaccination through week 20-8 post-revaccination (Figure 5). However, a significant increase was observed for the MLV/KV group on week 24-12 for the frequency of IFN-γ mRNA-positive CD335+ cells after BVDV-2a stimulation as compared to the KV/MLV and control groups (Figure 5). While not significantly different from the MLV/KV group, an increase in the frequency of IFN-γ mRNA-positive CD335+ cells after BVDV-2 stimulation was observed on week 12 post-vaccination through week 16-4 post-revaccination for the KV/MLV group (Figure 5).

As previously reported [17], a decrease in the frequency of BVDV positive PBMC after BVDV stimulation was observed in vaccinated cattle. After BVDV-2a stimulation, the frequency of BVDV positive PBMC was assessed (Figure 6). After BVDV-2a stimulation, a significant decrease in the frequency of BVDV-2a positive cells was observed for the MLV/KV on weeks 8 and 12 post-vaccination, and at all time points post-revaccination as compared to controls (Figure 6). No significant differences were observed between the KV/MLV and controls groups after BVDV-2a stimulation over the course of the study (Figure 6). A significant decrease in the frequency of BVDV-2a positive cells in the MLV/KV group compared to KV/MLV was observed at weeks 8 and 12 post-vaccination (Figure 6).

### 3.2. BVDV-1a Results

A similar trend was observed for the BVDV-1a data, as observed for the BVDV-2a data. The BVDV-1a is included for the comprehensiveness of the data rather than comparative purposes due to the different antigens used in vaccines and differing responses that may be elicited.

An initial increase in BVDV-1a titers was observed for the MLV/KV group on week 8 post-vaccination, but a significant increase was not observed until week 12 when compared to the control and KV/MLV groups (Appendix A). Post-revaccination on week 2 (14-2), BVDV-1a titers remained greater for the MLV/KV group, compared to the control and KV/MLV groups (Appendix A). Additionally, peak BVDV-1a titers for the MLV/KV group were observed on week 14-2. For the KV/MLV group, BVDV-1a titers significantly different from the control animals were not observed until week 4 (16-4) post-revaccination (Appendix A). While titers were not significantly different from the MLV/KV group, an increase in titer was observed for the KV/MLV group from week 16-4 through to the conclusion of the study. Interestingly, cattle in the KV/MLV group had the highest titer of 710.3 at the final sample time point (24-12) for BVDV-1a as compared to 373.3 for the MLV/KV group (Appendix A).

After in vitro stimulation of PBMC with BVDV-1a, a significant difference in CD25 MFI was observed for the MLV/KV group on weeks 8 and 12 post-vaccination compared to control and KV/MLV groups (Appendix A). Post-revaccination week 2 (14-2), increased CD25 MFI was only observed in BVDV-1a stimulated PBMC for the KV/MLV group. However, for the remainder of the timepoints, weeks 14-2 through 20-8 post-revaccination, CD25 MFI was significantly increased in both MLV/KV and KV/MLV as compared to the control animals (Appendix A). Additionally, the peak CD25 MFI for BVDV-1a stimulated PBMC for both the MLV/KV and KV/MLV groups was observed on week 20-8 (Appendix A). The greatest CD25 MFI for BVDV-1a stimulated PBMC was observed from cattle in the KV/MLV group on week 20-8 (Appendix A).

After BVDV-1a PBMC stimulation, a significant increase in frequency of IFN-γ mRNA-positive CD4+ cells for the MLV/KV group from 8–12 weeks post-vaccination compared to the control and KV/MLV groups was observed (Appendix A). Interestingly, post-revaccination, despite a trend towards an increase in the frequency of IFN-γ mRNA-positive CD4+ in the MLV/KV group, no significant differences were observed between the three vaccinate groups (Appendix A). Additionally, the peak frequency of IFN-γ mRNA-positive CD4+ cells after BVDV-1a stimulation was observed on week 16-4 post-revaccination, although no significant differences from the KV/ML and control group were observed.

After BVDV-1a PBMC stimulation, a significant increase in the frequency of IFN-γ mRNA-positive CD8+ cells was observed for the MLV/KV group on weeks 8 and 12 post-vaccination compared to the control and KV/MLV groups (Appendix A). Despite a trend towards an increase in the frequency of IFN-γ mRNA-positive CD8+ cells after BVDV-1a stimulation for the MLV/KV group through the conclusion of the study, no significant differences were observed until the last sample time point on week 24-12 between the MLV/KV group and controls (Appendix A). The peak frequency of IFN-γ mRNA-positive CD8+ cells after BVDV-1a stimulation was observed on week 16-4 post-revaccination, although no significant differences from the KV/MLV and control group were observed (Appendix A). No significant differences in the frequency of IFN-γ mRNA-positive CD8+ cells after BVDV-1a stimulation were observed for the KV/MLV group when compared to the MLV/KV and control groups (Appendix A).

After BVDV-1a PBMC stimulation, a significant increase was observed in the frequency of IFN-γ mRNA-positive CD335+ cells for the MLV/KV group on weeks 12 and 16-4 as compared to the control group (Appendix A). A significant increase was observed for the MLV/KV group on week 24-12 for the frequency of IFN-γ mRNA-positive CD335+ cells after BVDV-1a stimulation as compared to the KV/MLV and control groups (Appendix A).

After BVDV-1a stimulation, MLV/KV vaccinates had significantly lower frequencies of BVDV-1a positive cells on week 12 post-vaccination and weeks 14-2 and 20-8 post-revaccination as compared to the control animals (Appendix A). KV/MLV vaccinates only showed a significant decrease in the frequency of BVDV-1a positive cells on weeks 14-2 and 20-8 post-revaccination, as compared to the controls (Appendix A).

## 4. Discussion

The benefits of vaccination with MLV BVDV vaccines have been demonstrated in in vivo vaccination challenge studies, supporting improved protection against reproductive diseases using MLV vaccines [3]. Furthermore, other studies have suggested that using two doses of a MLV vaccine in heifers followed by a KV provided superior protection against BVDV persistent infection compared to the using all three doses of a MLV vaccine [6]. In contrast, using only a killed multivalent viral vaccine regimen resulted in an inferior and concerning lack of protection against BVDV challenges [24]. The increased protection provided by BVDV MLV vaccines is thought to be associated with a more robust CMI response as determined by cell-mediated recall responses measured by the production of IFN-γ and expression of CD25+ as markers for activated T-lymphocytes. Previous literature has suggested that both MLV and KV containing BVDV exhibit a recall memory response as observed by inducing increased expression of IFN-γ in T-cell subsets after re-stimulation with BVDV in vitro [25,26,27]. It should be noted that while it has been suggested that inactivated BVDV vaccines induce a CMI response, these results are discordant with the in vivo data that support that MLV vaccines induce a more effective response. The data from this study support that BVDV MLV vaccines are associated with a more robust CMI response as demonstrated by the increased frequency of IFN-γ (mRNA)-expressing T cells as well as an increase in MFI CD25+ expression after cattle received the MLV vaccine. Additionally, IFN-γ has been demonstrated to have antiviral properties as previously reported [28], and the results from the current study corroborate these findings as cattle with increased IFN-γ, after administration of the MLV vaccine, had a lower frequency of BVDV positive cells. While a decrease in BVDV positive cells has previously been reported in vitro [17], in vivo vaccination studies also support this observation as cattle that have been vaccinated with a BVDV MLV vaccine have fewer days of viremia after challenge as compared to cattle vaccinated with a KV [6,24].

More recently studies have demonstrated the importance of early life influenza exposures in shaping long-term influenza-specific immunological memory, and this is termed immunological imprinting [29]. Specifically, the data would support that not only does initial exposure to influenza influence antigenic responses, but natural infection or vaccination with an inactivated antigen differentially primes influenza-specific CD4 T-cell responses [29]. Influenza infection primed a CD4 T cell response of greater magnitude as compared to the inactivated influenza vaccine after initial exposure. Furthermore, this trend remained when vaccination and infection cohort groups were subsequently all administered an inactivated vaccine after either the initial exposure from infection or vaccination [29]. This highlights the importance of how initial exposure to an antigen can have implications not only from an antigenic standpoint, but also from a priming or imprinting perspective. Therefore, consideration should be given when selecting either a MLV or KV vaccine for the first exposure to the virus. The data from the current study would also suggest that the initial presentation of the virus using a MLV or KV may have primed the immune response differently. Similar to the influenza data, the magnitude of T-cell responses, as determined by recall measures of IFN-γ and CD25+, was greater when initial exposure was to the MLV vaccine.

Interestingly, the VNT were higher at the end of the study for the cattle that received the KV vaccine initially. This potentially further illustrates the importance of how the initial presentation of the antigen primes the immune response and contributes changes in the immune profile/responses such as predominates as a Th1 or Th2 phenotype. Collectively, these findings support the importance of initial antigen exposure and the functionality of T cell and, in general, immune responses. Therefore, considerations regarding the timing of vaccination and the types of vaccine used for vaccination deserve particular attention and need further characterization, and a number of variables should be considered when selecting the most appropriate vaccination regime. It should be noted that typically MLV vaccines are multivalent and contain other viral fractions, namely BHV-1. BHV-1 has been demonstrated to be associated with reproductive failure in cattle leading to safety concerns [5]. Therefore, a balance between protection and undesirable effects such as safety concerns with MLV vaccines need to be considered to determine the most appropriate vaccine regime. Lastly, it should be noted that while IFN-γ and CD25+ were greater for cattle vaccinated with the MLV, measurable IFN-γ and CD25+ responses were detected after subsequent administration of the MLV vaccine in the cattle vaccinated with KV initially. It is unknown if there is a specific IFN-γ response or expression of CD25 that must be achieved and is associated with protection. Another consideration is the notoriously costly process of an immune response that often results in trade-offs with other costly activities such as reproduction and growth [30,31,32,33]. While not the focus of the current study, this does highlight that a robust and greater immune response may not necessarily benefit production animals and may not be necessary to confer protection. A better understanding of leveraging different vaccination strategies to maximize production, mitigate safety concerns, and prime immune responses is needed.

## Figures and Tables

**Figure 1 viruses-15-00703-f001:**
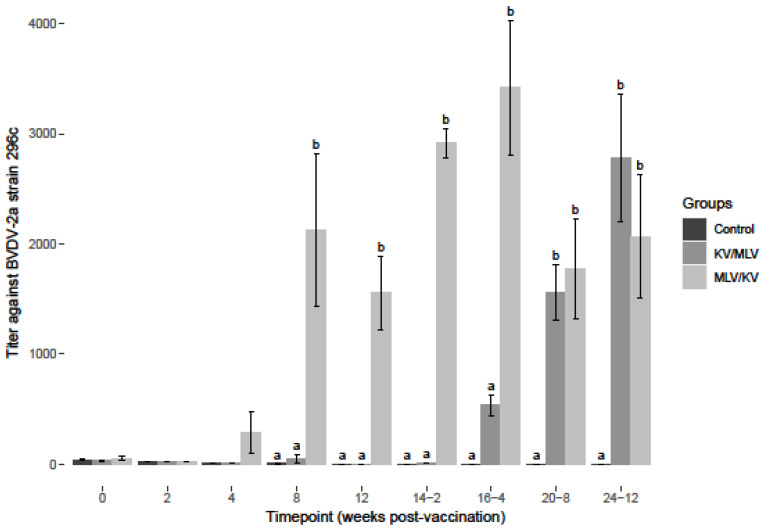
Average virus neutralization titers from sera collected at respective timepoints over the course of the study against BVDV-2a (296c) refence strain. Values at each timepoint, not connected by the same letter are significantly different (*p* < 0.05).

**Figure 2 viruses-15-00703-f002:**
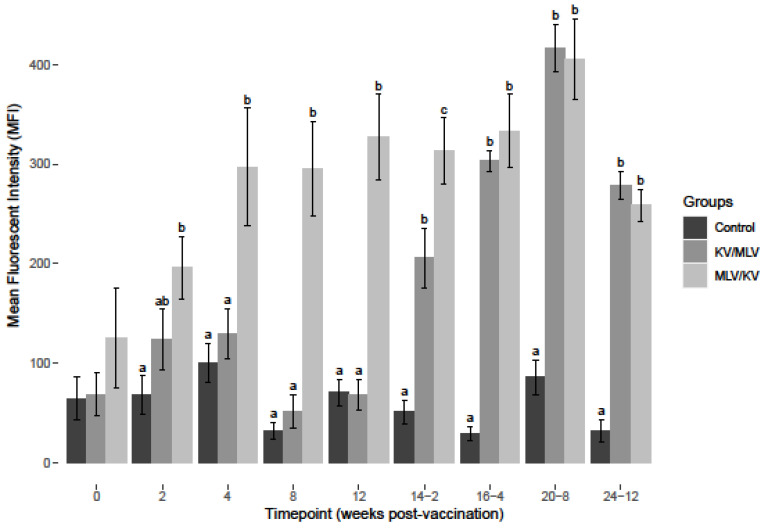
Average mean fluorescent intensity (MFI) for CD25^+^ (IL-2α receptor) PBMC after 24-h stimulation at each respective timepoint over the course of the study with BVDV-2a (PI-28). Values at each timepoint, not connected by the same letter are significantly different (*p* < 0.05).

**Figure 3 viruses-15-00703-f003:**
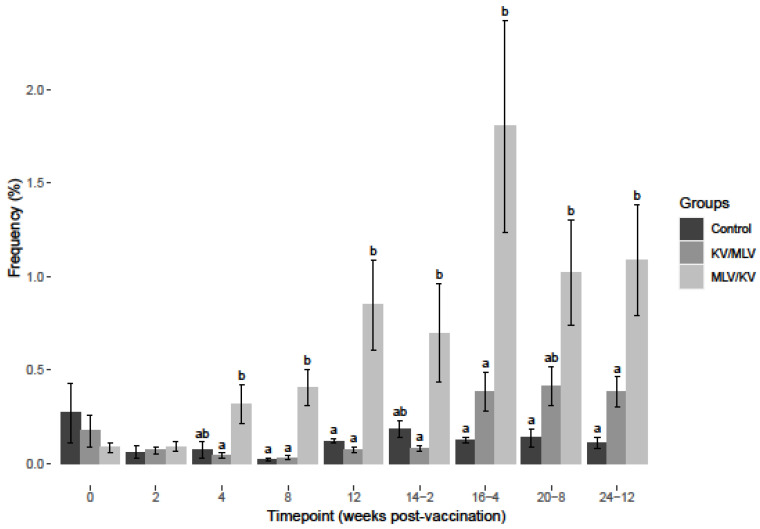
Frequency of IFN-γ mRNA-positive CD4^+^ cells after 24-h stimulation at each respective timepoint over the course of the study with BVDV-2a (PI-28). Values at each timepoint, not connected by the same letter are significantly different (*p* < 0.05).

**Figure 4 viruses-15-00703-f004:**
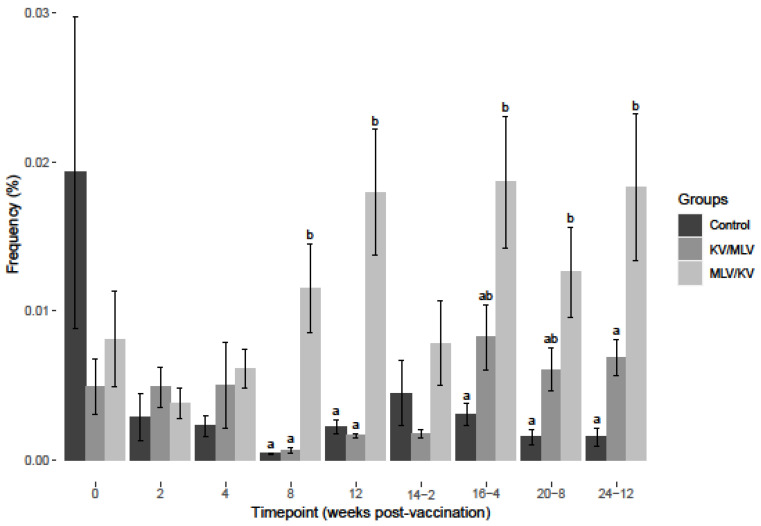
Frequency of IFN-γ mRNA-positive CD8^+^ cells after 24-h stimulation at each respective timepoint over the course of the study with (a) BVDV-1a (PI-34) and (b) BVDV-2a (PI-28). Values at each timepoint, not connected by the same letter are significantly different (*p* < 0.05).

**Figure 5 viruses-15-00703-f005:**
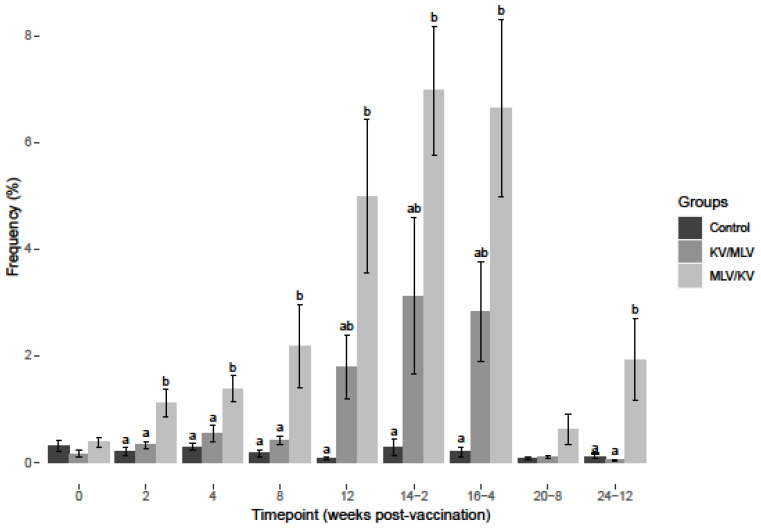
Frequency of IFN-γ mRNA-positive CD335^+^ cells after 24-h stimulation at each respective timepoint over the course of the study with BVDV-2a (PI-28). Values at each timepoint, not connected by the same letter are significantly different (*p* < 0.05).

**Figure 6 viruses-15-00703-f006:**
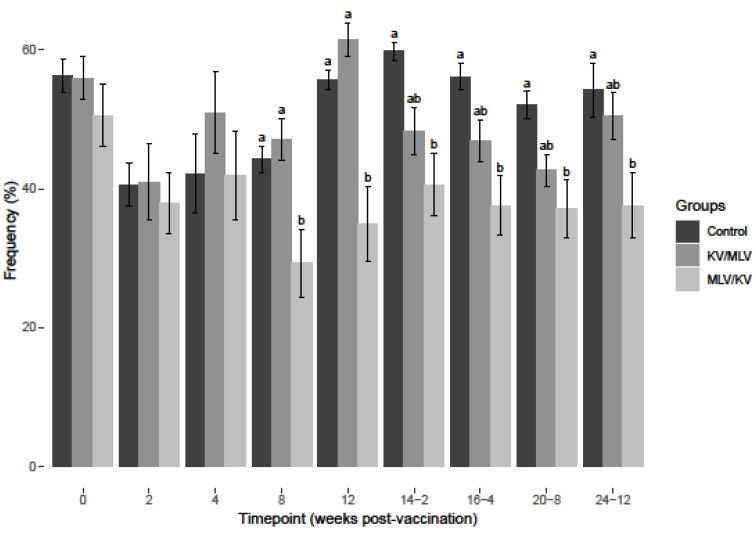
Frequency of PBMC positive for the respective BVDV PrimeFlow probe after 24-h stimulation at each respective timepoint over the course of the study with BVDV-2a (PI-28). Values at each timepoint, not connected by the same letter are significantly different (*p* < 0.05).

**Table 1 viruses-15-00703-t001:** Primary and secondary antibodies used for surface marker expression on PBMC’s and Primeflow probes used for cell mediated immune response comparisons.

Antibody and Probes	Cell Marker	Clone	Isotype	Fluorochromes
* CD2	T and NK cells	MUC2A	IgG2a	BV421
CD25	IL-2 receptor/activation	LCTB2A	IgG3	BUV395
CD335	NK cells	ASK1	IgG1	BV711
CD4 PrimeFlow probe	T cell subset			AF568
CD8α PrimeFlow probe	T cell subset			AF488
IFN-γ PrimeFlow probe	IFN-γ mRNA/stimulation			AF750
** BVDV PrimeFlow probe	BVDV viral RNA			AF647

* Cluster of differentiation (CD); ** BVDV PrimeFlow probe designed for each respective BVDV strain (PI-34 or PI-28).

## Data Availability

The data presented in this study are available on request from the corresponding author.

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
