# Peer review of "Response to Bovine Viral Diarrhea Virus in Heifers Vaccinated with a Combination of Multivalent Modified Live and Inactivated Viral Vaccines"

_viruses, 2023, doi:10.3390/v15030703_

Round 1

Reviewer 1 Report

               Vaccine protocols for BVDV are highly variable depending on the type of vaccine – killed or modified live, the type of operation – dairy, feedlot, open range beef, perceived health risks, veterinarian experience, and level of understanding of the problem by the owner.  Most any vaccine administered properly will largely prevent acute disease, but modified live vaccines seem to be superior in preventing fetal protection.  Some studies have suggested that mixing of vaccine types might give enhanced protection.  This study is attempting to evaluate whether this might be the case by measuring humoral and CMI responses in animals immunized in two different patterns – killed then MLV and MLV then killed.  My main concern for this study is that the vaccine protocols used are not those that are currently used or those that have shown promise of superiority.  The authors do not discuss their rationale for initial vaccination followed by a boost 12 days later.  Even with killed vaccines, the minimum interval between initial dose and booster dose is 3 weeks with evidence that longer intervals establish a more robust response. The data generated in this study may well be valid, but does it apply to the more standard vaccine protocols being used?  The authors need to fully justify their selection for the study time intervals.

               My second major issue is the presentation of the data largely as a comparison of the responses of the animals to BVDV-1a vs BVDV-2a.  It could be argued that any differences seen are due to the vaccine formulations and if one used different products, one might see something different.  I do not think that the authors can justify highlighting differences noted between 1a and 2a responses. The focus should be totally on the differences seen between the vaccine groups using one of the antigens. The rationale for the study was to assess vaccine protocols not the antigens used.  Accordingly, I would recommend eliminating the 1a graphs from the figures giving more landscape for 2a responses – more distinct differences among the graph bars and larger lettering. 

               The Vaccine group labels in the all the graphs are confusing – one has Controls, KV/MLV, and MLV/KV, not just KV and MLV.  Also, what do the letters represent on the bars at the various time points?  The text notes that some time point differences are statistically different.  That should be noted on the bar graphs.  If one eliminates the 1a data, then the 2a figures can be made larger and more readable.  Figure legends should also contain more information.

               As a minor issue, the description for the growing of the viruses is way too detailed as there is nothing unique about how the authors grow BVDV

The two step (killed-MLV) protocol was championed in Germany with multiple papers by Frey et al.  some of these should be referenced.

Preventive Veterinary Medicine 72 (2005) 109–114    Implementation of two-step vaccination in the control of bovine viral diarrhoea (BVD)V. Moennig , K. Eicken , U. Flebbe  H.-R. Frey,  B. Grummer , L. Haas,  I. Greiser-Wilke , B. Liess

Reviewer 2 Report

The manuscript of Shollie M. Falkenberg et al. Response to bovine viral diarrhea virus in heifers vaccinated with a combination of multivalent modified live and inactivated viral vaccinesThe present study is focused on comparing vaccination methods KV/MLV and MLV/KV. The author found that differences in virus neutralizing titers and cell mediated immune were observed between KV/MLV and MLV/KV. There are some issues that need to be resolved.

1. The describe of Frequency of IFN-γ mRNA positive CD4+, CD8+, and CD335+ populations, as well as increased mean fluorescent intensity of CD25+ cells was increased for the MLV/KV heifers as compared to KV/MLV and controls is simple compared to the results in the study.

2. The quality of all images is poor in the article. The authors should improve the clarity of the figure.

3. The discussion part revealed some the problems. The authors should add more critical analysis combined with the results of this study.

4. The author needs to further analyze the differences, advantages and disadvantages of the two immunization methods.

5. The authors need to analyze the importance of switching immunization methods from the results of the study.

Round 2

Reviewer 1 Report

Presentation of the data has been greatly improved with the separation of the BVDV strain data.  At the end of the day, I am not sure that the best vaccination program for any management system can be built on these data.

Minor points:

Legend to Fig 4 needs to be corrected - no 1a data

Same figure submitted for all supplemental figures.

Line 444 needs to be changed.

Line 248 should be reference to Supp Fig 1.

Line 246 - "or" to "for"

Line 219 - "isolated" to "isolates.
